# Focus on ROS1-Positive Non-Small Cell Lung Cancer (NSCLC): Crizotinib, Resistance Mechanisms and the Newer Generation of Targeted Therapies

**DOI:** 10.3390/cancers12113293

**Published:** 2020-11-06

**Authors:** Alberto D’Angelo, Navid Sobhani, Robert Chapman, Stefan Bagby, Carlotta Bortoletti, Mirko Traversini, Katia Ferrari, Luca Voltolini, Jacob Darlow, Giandomenico Roviello

**Affiliations:** 1Department of Biology and Biochemistry, University of Bath, Bath BA2 7AY, UK; bsssb@bath.ac.uk (S.B.); jbd31@bath.ac.uk (J.D.); 2Section of Epidemiology and Population Science, Department of Medicine, Baylor College of Medicine, Houston, TX 77030, USA; navidsobhani19@gmail.com; 3University College London Hospitals NHS Foundation Trust, 235 Euston Rd, London NW1 2BU, UK; robert.chapman2@nhs.net; 4Department of Dermatology, University of Padova, via Vincenzo Gallucci 4, 35121 Padova, Italy; bortoletti.carlotta@gmail.com; 5Unità Operativa Anatomia Patologica, Ospedale Maggiore Carlo Alberto Pizzardi, AUSL Bologna, Largo Bartolo Nigrisoli 2, 40100 Bologna, Italy; traversini.mirko@gmail.com; 6Respiratory Medicine, Careggi University Hospital, 50139 Florence, Italy; katia3ferrari@gmail.com; 7Thoracic Surgery Unit, Careggi University Hospital, Largo Brambilla, 1, 50134 Florence, Italy; luca.voltolini@unifi.it; 8Department of Health Sciences, Section of Clinical Pharmacology and Oncology, University of Florence, Viale Pieraccini, 6, 50139 Florence, Italy; giandomenicoroviello@gmail.com

**Keywords:** lung cancer, solid tumours, crizotinib, toxicity, TKI inhibitors, resistance mechanisms, targeted therapies, ROS1, NSCLC

## Abstract

**Simple Summary:**

Genetic rearrangements of the *ROS1* gene account for up to 2% of NSCLC patients who sometimes develop brain metastasis, resulting in poor prognosis. This review discusses the tyrosine kinase inhibitor crizotinib plus updates and preliminary results with the newer generation of tyrosine kinase inhibitors, which have been specifically conceived to overcome crizotinib resistance, including brigatinib, cabozantinib, ceritinib, entrectinib, lorlatinib and repotrectinib. After introducing each agent’s properties, we provide suggestions on the best approaches to identify resistance mechanisms at an early stage, and we speculate on the most appropriate second-line therapies for patients who reported disease progression following crizotinib administration.

**Abstract:**

The treatment of patients affected by non-small cell lung cancer (NSCLC) has been revolutionised by the discovery of druggable mutations. ROS1 (c-ros oncogene) is one gene with druggable mutations in NSCLC. ROS1 is currently targeted by several specific tyrosine kinase inhibitors (TKIs), but only two of these, crizotinib and entrectinib, have received Food and Drug Administration (FDA) approval. Crizotinib is a low molecular weight, orally available TKI that inhibits ROS1, MET and ALK and is considered the gold standard first-line treatment with demonstrated significant activity for lung cancers harbouring ROS1 gene rearrangements. However, crizotinib resistance often occurs, making the treatment of ROS1-positive lung cancers more challenging. A great effort has been undertaken to identify a new generation or ROS1 inhibitors. In this review, we briefly introduce the biology and role of ROS1 in lung cancer and discuss the underlying acquired mechanisms of resistance to crizotinib and the promising new agents able to overcome resistance mechanisms and offer alternative efficient therapies.

## 1. Introduction

Lung cancer is a leading cause of cancer mortality in both men and women worldwide, despite major recent therapeutic breakthroughs in the field [1]. Non-small cell lung cancer (NSCLC) accounts for the vast majority (85%) of lung malignancies. NSCLCs can be divided by their histological subtypes such as squamous cell carcinoma, large cell carcinoma and adenocarcinoma, with the latter the most diagnosed histological subtype. Two out of three advanced NSCLC patients harbour driver genetic alterations, including *KRAS*, *BRAF*, *EGFR*, *MET* and *ERBB2* mutations or *ALK*, *ROS1*, *RET* and *NTRK* genomic rearrangements [2]. Currently, the vast majority of these alterations are therapeutically targetable, with *KRAS* the most commonly altered gene in NSCLC (approximately 13%) and currently being targeted in a phase I clinical trial with sotorasib, a low molecular weight, highly selective KRAS^G12C^ inhibitor [3]. The recent administration of TKIs as a first-line regimen for *ALK*-rearranged and *EGFR*-mutated NSCLC outperformed the platinum-base chemotherapy in terms of response rates, progression-free survival (PFS) and overall survival (OS) [4]. Malignancies harbouring *ROS1* gene rearrangements can be targeted by TKIs. Crizotinib, a tyrosine kinase inhibitor of ALK, ROS1 and cMET, is the first agent to report clinical efficacy in this subgroup of patients according to preclinical, phase I, retrospective and prospective studies [5,6,7]. Unfortunately, following initial positive responses to crizotinib, a large number of *ROS1*-rearranged patients show progression of disease (sometimes with brain metastasis) due to the occurrence of resistance mechanisms, such as the bypass of signalling pathways and novel mutations affecting the kinase domains of ROS1 [8]. However, given the relatively long survival of ROS1-positive NSCLC patients, a more recent ROS1 inhibitor generation is currently under investigation, with an encouraging improvement in the control of central nervous system (CNS) metastasis and a demonstrated efficacy in both pretreated and naïve crizotinib patients [9].

Here we review the current research involving *ROS1*-rearranged NSCLC patients, including *ROS1* discovery, diagnostics, mechanisms exerted by malignant cells to evade inhibition and strategies to overcome acquired resistance. In addition, we focus on clinical efforts to target altered-ROS1 with standard regimens (crizotinib) and the more recent generation of ROS1 TKIs.

## 2. ROS1 Discovery and Signalling Pathway

ROS proto-oncogene 1 (ROS1), encoded by *ROS1* gene located at chromosome 6q22.1, belongs to the subfamily of tyrosine kinase insulin receptors and was first identified in 1986 during research involving the chicken sarcoma RNA UR2 tumour virus [10]. The physiological role of ROS1 is still controversial; its expression has been observed mainly in lung tissue, followed by the cervix and colon. Tyrosine kinase insulin receptors conceivably play a key role in embryonic development [11]. ROS1 protein shows substantial homology to ALK (both belong to insulin receptor superfamily), particularly within the ATP binding site (84% homology) and the kinase domains (64% homology) [5,12]. Genomic alteration of *ROS1* is well known and normally leads to gene fusion with several fusion partners (Table 1) [2,5,11,13,14,15,16,17,18,19,20,21,22,23]; the resulting fusion proteins are robust oncogenic drivers [24]. As a consequence, ROS1 kinase activity is constitutively activated, leading to increased cell proliferation, survival and migration due to the upregulation of JAK/STAT, PI3K/AKT and MAPK/ERK signalling pathways [11]. ROS1 has shown tumorigenic potential in vitro and in vivo, with glioblastoma the first human cancer shown to harbour ROS1 rearrangements [10]. ROS1 fusions have subsequently been observed in other malignancies including NSCLC [13], melanoma [19] and occasionally cholangiocarcinoma [25], angiosarcoma [26], ovarian [27], gastric [28] and colorectal cancer [29]. *ROS1* alterations can neither be inherited nor acquired as genetic alteration [30].

## 3. ROS1 in Lung Cancer and Brain Metastasis

First described in 2007 [13], *ROS1* rearrangements account for 1–2% of NSCLC patients with an estimated 10,000–15,000 new cases every year worldwide (Table 1) [24,30]. Patients are often light smokers or non-smokers, and tend to be young. Adenocarcinoma is the most common histological subtype identified, but large-cell and squamous histology have also been reported [16,31]. *ROS1*-positive cancers show overlapping clinical features with *ALK*-rearranged NSCLC, although genetic rearrangements in *ROS1*, *EGFR* and *ALK* tend to be mutually exclusive [32]. Additionally, some ROS1-positive cancers demonstrate a second driver mutation, mainly *KRAS* and *EGFR*, after initial treatment with a TKI [16,33,34]. In clinical practice, ROS1-positive patients responded better to chemotherapy (CT) than NSCLC patients harbouring other driver mutations, with an estimated 60% objective response rate (ORR), 89.5% disease control rate (DCR) and a 7 month PFS. The better response described is likely due to the sensitivity of *ROS1*-positive cancers to pemetrexed (Alimta) [14,35,36,37,38]. It has been speculated that this significant pemetrexed susceptibility might arise from the limited number of thymidylate synthetase (TS) transcripts in *ROS1*-positive patients [39,40,41,42]. Additionally, the response of ROS1-positive cancers to immunotherapy appears to be infrequent [43]. Furthermore, receptor tyrosine kinases (RTKs) found in *ROS1*-positive cancers are druggable targets and TKIs (such as crizotinib) therefore emerged as promising agents for the treatment of these cancers [8]. TKIs do not target fast-growing cells (as CT agents normally do), but only cells harbouring specific genomic alterations, and as a result, they lead to fewer side effects than CT agents. However, the development of resistance mechanisms to TKIs can be expected [5,30].

### Brain Metastasis

Metastases affecting the CNS are commonly observed in patients with lung cancer and *ROS1*-positive cancer patients are no exception. One study observed that approximately 40% of stage IV *ROS1*-positive lung cancer patients were diagnosed with brain metastasis [44]. Crizotinib is not the preferred agent for patients with brain metastasis due to its inability to penetrate the blood–brain barrier (BBB) [45]. Newer TKIs including repotrecninib, ceritinib, lorlatinib and entrectinib have been developed to penetrate the BBB. Entrectinib is the only agent approved by the FDA for NSCLC patients harbouring *ROS1* alterations and is currently the preferred agent in those with brain metastasis [46,47]. Despite the encouraging results with entrectinib, radiation therapy is the preferred technique to treat brain metastasis; *ROS1*-rearranged cancers have shown significant sensitivity to this approach [8,48]. When multiple brain metastases are diagnosed, whole-brain radiation is the common approach. Alternatively, when fewer lesions are observed (three or four), stereotactic radiation, which includes gammaknife, cyberknife and linear accelerator (LINAC)-based radiotherapy approaches, is preferred; this results in fewer side effects and better control of potential in situ relapse when coupled with whole brain irradiation [49,50,51,52]. The full list of current clinical trials using TKI against *ROS1*-rearranged lung cancer is reported in Table 2.

## 4. ROS1 Rearrangement Diagnosis

*ROS1*-positive patients account for 1–2% of all lung cancer diagnoses, but the clinicopathological features alone are not robust enough for their detection. As a result, different techniques are employed to identify *ROS1* rearrangements, including immunohistochemistry (IHC), fluorescence in-situ hybridisation (FISH), reverse transcriptase-polymerase chain reaction (RT-PCR) and next generation sequencing (NGS) (Table 3) [53]. Immunohistochemistry is considered an efficient tool to diagnose ROS1 rearrangements. IHC has the benefit of low cost, high sensitivity (ranging from 90% to 100%), low operator requirements and shorter turnaround times when compared to FISH [50,54,55,56,57]. However, subjective evaluation, specificity and sensitivity of antibodies and tissue fixation might negatively influence results [58].

Fluorescence in-situ hybridisation has traditionally been considered the “gold standard” technique for the detection of *ROS1* gene fusions, mainly due to the use of identical tests for *ALK* rearrangements [15,50,59]. In FISH, the specimen is deemed “positive” when more than 15% of evaluated cells, or more than 50 malignant cells [60], show single 3′ (centromeric) or split 3′ and 5′ (telomeric) signal [55,56]. FISH is considered technically challenging, moderately expensive, labour intensive and prone to false-negative results [58].

Reverse transcriptase-polymerase chain reaction (RT-PCR) requires an RNA sample to be converted to complementary DNA (cDNA) first and then amplified, allowing specific identification of gene fusion partners. RT-PCR sensitivity and specificity are good, but it relies on RNA obtained from formalin-fixed and paraffin-embedded (FFPE) samples, which might limit its use in routine lab work [61]. An emerging test is currently provided by NanoString, which gives 100% sensitivity and specificity for *ROS1* rearrangement identification [62]. However, this PCR method can introduce sequence artefacts leading to potential errors when processing highly fragmented or short nucleic fragments [61]. Lastly, next generation sequencing (NGS) enables the identification, in parallel, of known or novel *ROS1* fusion alterations (and other oncogenic mutations) using either DNA or RNA as a starting material [63]. Moreover, DNA-based NGS can also detect fusion gene rearrangements with short intron regions, though it might struggle with long intron regions, and often fusion genes might differ from the RNA-messenger fusions [64]. It is highly possible that NGS will become the gold standard test for the identification of ROS1 fusion genes in the foreseeable future, due to its capability for massively parallel sequence testing and the detection of both known and novel gene rearrangements. Nonetheless, exhaustion of specimen and extended turnaround time might hamper its introduction in routine clinical practice [4]. The full list of the advantages and disadvantages of each technique is reported in Table 2.

## 5. Targeting the ROS1 Oncogene with Crizotinib

Crizotinib (Xalkori), originally formulated as an antihepatocyte growth factor receptor (MET) agent, is a TKI against several targets including ROS1, anaplastic lymphoma kinase (ALK) and MET [31,65]. Crizotinib inhibits ATP-dependent cellular functions by binding to respective protein kinase domains leading to potent ROS1, MET and ALK suppression [11]. Its efficacy was firstly reported in a ROS1-rearranged cell line when the inhibition of ROS1 phosphorylation led to a robust dose-dependent (IC50 of 60 nM) cell apoptosis; IC50 of 8 nM and 40–60 nM against MET and ALK respectively [24,65,66,67]. However, it was only in 2012 that the first report on the use of crizotinib in the treatment of *ROS1*-rearranged lung cancers was published. A young, non-smoking male diagnosed with *ALK* and *EGFR*-negative multifocal NSCLC showed disease progression after first-line erlotinib and developed genomic alteration in *ROS1*. The patient was treated with 250 mg crizotinib twice per day, which led to a significant improvement in patient-reported symptoms within 7 days and dramatic shrinkage of malignant lesions at imaging within 8 weeks [31].

In the phase I PROFILE 1001 clinical trial, 50 patients with *ROS1*-rearranged NSCLC were included (as an expansion cohort) alongside *ALK*-rearranged advance NCLSC patients, all of whom were treated with crizotinib [5,68]. *ROS1*-positive patients reported a DCR of 90%, a median PFS of 19.2 months, an ORR of 72% and an OS rate of approximately 85% at 12 months. 5 patients (10%) showed treatment-related adverse events (AEs) including vomiting, nausea, constipation, diarrhoea, fatigue, visual impairment and dizziness; the vast majority evaluated as clinically manageable with only 4% evaluated as grade 3 of 4, and no grade 5 AEs were observed [5]. Based on this study, in 2016 the FDA and EMA approved the use of crizotinib for advanced *ROS1*-rearranged NSCLC patients [69]. Subsequently, in 2019 the updated results of the study showed a final median PFS and OS of 19.3 months and 51.4 months respectively [68]. Following these results, prospective and retrospective studies were carried out to confirm crizotinib efficacy in *ROS1*-rearranged NSCLC. The EUROS1 Cohort study retrospectively evaluated crizotinib efficacy in 32 *ROS1*-positive patients and reported a disease control rate of nearly 87%, a PFS of 44% at 12 months and, remarkably, 5 patients achieved a complete response [14]. In the Acsè prospective phase II trial, the 39 *ROS1*-positive NSCLC patients included showed a DCR of 89% and an overall ORR of 54% with 43% of patients evaluated to have no-progression of disease after 12 months [70]. Among 34 patients recruited in the EUCROSS phase II trial, an ORR of approximately 70% was observed [71]. The Asiatic side, a prospective study recruiting 127 patients, reported a DCR of 89% at 8 weeks, an ORR of 72%, a median PFS and median OS of 15.9 and 32 months respectively, with 14 patients (11%) achieving a complete response [6]. The overlapping clinical responses between European and Asian studies demonstrated the efficacy of crizotinib across different ethnicities.

## 6. Mechanisms of Resistance to Crizotinib

The development of crizotinib resistance in previously sensitive cancer cells is a key cause of treatment failure and invariably leads to disease progression. Proposed mechanisms of crizotinib resistance in cancer cells can be broadly split into either the genetic alteration of the drug target or the activation of other signalling pathways (e.g., bypass signalling; Figure 1) [72,73]. Studies suggest that point mutations within the ROS1 kinase domain can lead to significant alterations of the drug target resulting in acquired resistance to crizotinib; such mutations decrease the potency of kinase inhibition [73,74]. Many examples of point mutations in the *ROS1* gene have been identified in crizotinib-resistant forms of NSCLC (Table 4). A *ROS1* D2033N mutation was detected in a patient with disease progression after a previous crizotinib-sensitive NSCLC CD74-ROS1 fusion diagnosis [75]. The D2033N mutation generates an aspartic acid to asparagine substitution at the kinase hinge region of ROS1 (within the ATP-binding site), leading to strong crizotinib resistance in vitro [30,75]. Similarly, G2032R is a mutation in the ROS1 kinase domain, which has been detected in samples of malignant cells post-crizotinib treatment, with the mutation not present at the pretreatment stage [76]. G2032R is thought to be the most common driving mutation of crizotinib resistance and the most difficult to treat from a pharmacological point of view [8,77]. Other examples include L2026M, which induces resistance to crizotinib by altering the gatekeeper position of the inhibitor binding pocket of ROS1 [73,78]. Additionally, L2155S is thought to confer crizotinib resistance via protein malfunction, whereas the S1986F/Y substitution within the kinase domain leads to an obstruction of the key site for activation, thereby increasing kinase activity [79].

Furthermore, the activation of bypass signalling as a method of crizotinib resistance in *ROS1* cancer cells has been described in various studies and identified in approximately 45% of crizotinib-resistant *ROS1*-rearranged NSCLC malignancies (Figure 1) [80,81]. The activation of the EGFR pathway has been shown to confer crizotinib resistance by reducing the dependence on ROS1 activity and increasing dependence on EGFR activity [79,82,83]. This signalling switch might lead *ROS1*-rearranged NSCLCs to be targeted in the future with EGFR inhibitors such as erlotinib and gefitinib [8]. Similarly, it has been shown that KIT pathway activation can lead to crizotinib resistance as well as promoting autophosphorylation and cell proliferation in vitro [72,81]. Consequently, it might be thought that the addition of a KIT inhibitor such as ponatinib might be beneficial in such cases. Lastly, upregulation of MAPK signalling plus the amplification of *TP53* and *HER2* have been reported in crizotinib-resistant cells and might play a key role in bypass mechanisms [45,80].

## 7. A New Generation of TKIs to Overcome Crizotinib Resistance and Combination Regimens

A novel and more potent generation of TKIs against genetic *ROS1*-alterations, which confer resistance to crizotinib, are currently under clinical investigation. These include brigatinib, cabozantinib, ceritinib, entrectinib, lorlatinib and repotrectinib. These newer agents have demonstrated clinical efficacy against *ROS1*-positive NSCLCs and more robust CNS penetration [8,76], and may be effective in the case of resistance resulting from a novel *ROS1* mutation [4]. Nonetheless, when the resistance results from the activation of bypass mechanisms (i.e., EGFR or KIT pathways), inhibition is still possible by targeting both ROS1 and the bypass pathway with the use of a combination therapy such as crizotinib plus an EGFR/KIT inhibitor [72].

### 7.1. Brigatinib

The ALK inhibitor brigatinib has demonstrated clinical activity against different mutations correlated with resistance to crizotinib, including *ROS1* gene rearrangements [84,85]. In three *ROS1* patients, recruited as an expansion arm of a larger phase II study that included 137 NSCLC patients, two (66%) had an objective treatment response. Moreover, one patient was crizotinib-naïve and reported partial response while the other patient was crizotinib pretreated and reported stable disease. Overall, brigatinib toxicity was reported to be manageable with increased lipase (9%) as primary grade 3 or 4 AE, while serious treatment-related AEs such as dyspnoea and pneumonia occurred in 5% of patients [86]. However, brigatinib did not show efficacy for patients harbouring G2032R, the commonest mutation in crizotinib resistance [12,87].

### 7.2. Cabozantinib

The multi-kinase inhibitor cabozantinib, which has demonstrated activity against *ALX*, *KIT*, *MET*, *RET*, *ROS1*, *VEGFR2* and *TIE2* [88], and is already approved for the treatment of medullary thyroid and renal cell carcinoma [89,90,91], has been administered by Drilon et al. in a phase II study enrolling lung cancer patients [75]. Within the aforementioned clinical trial, a daily dose of 60 mg cabozantinib showed efficacy (in terms of imaging and symptoms) in a young female non-smoker harbouring D2033N-*ROS1* genomic alteration who acquired resistance to crizotinib. Additionally, Sun et al. reported clinical efficacy of cabozantinib in four *ROS1*-positive NSCLC patients with resistance to crizotinib and ceritinib; the PFS of patients administered with cabozantinib ranged from 4.9 to 13.8 months with neutropenia, xeroderma and pulmonary embolism the primary AEs reported [92]. Although cabozantinib has demonstrated its ability to overcome resistance driven by newer identified secondary mutations, it does have a challenging side effect profile.

### 7.3. Ceritinib

Ceritinib, a highly specific and robust ROS1 inhibitor, showed 20 times higher efficacy than crizotinib in a rat model [93], with a satisfactory BBB penetration ratio [94,95,96,97]. Ceritinib was approved after a phase II study, which included 32 *ROS1*-positive NSCLC patients administered with a daily dose of 750 mg. Among crizotinib-naïve and crizotinib pretreated patients, the overall ORR was 62% with a median PFS of 9.3 months. Among 8 patients with CNS disease, 63% reported disease control, while 25% showed an intracranial response [93]. However, patients receiving ceritinib reported critical rates of toxicities, primarily diarrhoea (78%), nausea (59%) and anorexia (56%) with grade 3 or higher AEs reported by 12 patients (37%) [93]. Clinical trials investigating different ceritinib combinations to mitigate toxicity are underway. Notably, ceritinib efficacy is restricted to crizotinib-naïve patients; it did not show efficacy towards crizotinib-resistant malignancies [95].

### 7.4. Entrectinib

Entrectinib (ROZLYTREK, Genentech Inc.) is an orally available, low molecular weight TKI, specifically designed to penetrate the BBB. Entrectinib received FDA approval for lung cancers harbouring *ROS1* alteration in 2019 [46,47,98], having demonstrated robust ALK and ROS1 inhibition, plus effective TRKA, TRKB and TRKC signalling suppression [99]. However, entrectinib has limited efficacy against G2032R, D2033N and L2026M *ROS1* mutants [87,100]. Entrectinib led to a median PFS of 19.0 months and an ORR of 77.4% within 53 *ROS1*-positive NSCLC patients recruited for a phase I/II trial [98,101,102]. Among those patients with brain metastasis, the intracranial ORR was 55% and median PFS of 12.9 months, which makes entrectinib advantageous compared to crizotinib in this cohort; intracranial progression on treatment occurred in three patients (15%). Entrectinib’s overall toxicity was manageable (primarily grade 1) and reversible with 27% and 4% of patients requiring dose reduction and treatment discontinuation, respectively. Main entrectinib-related toxicities reported were dysgeusia (41%), fatigue (28%) and dizziness (25%) [102].

### 7.5. Lorlatinib

Lorlatinib is a potent oral inhibitor of ALK (FDA approval in 2017) and ROS1 with activity against several *ROS1* mutants (G2032R, D2033N and S1986Y), which confer resistance to crizotinib and ceritinib, and the ability to penetrate the BBB [78,103,104]. Promising results have recently been reported by a phase II study that included 47 *ROS1*-positive NSCLC patients, including both TKI-naïve and pretreated patients [104,105]. A median PFS of 21.0 months and an ORR of 61.5% were observed among the 13 crizotinib-naïve patients while a median PFS of 8.5 months and an ORR of 26.5% was reported among the 34 patients pretreated with crizotinib. Surprisingly, an ORR of 66.7% was observed among the 6 patients with brain metastasis who received no prior treatment [104]. The overall main AEs reported were hypercholesterolemia, hypertriglyceridemia and oedema. Of note, lorlatinib was reported to be efficacious towards resistance mechanisms due to bypass signalling activation, although it showed limited efficacy in patients harbouring G2032R [105].

### 7.6. Repotrectinib

Repotrectinib (TPX-0005) is a low molecular weight, potent TKI with demonstrated efficacy against TRK receptors, *ALK* and most *ROS1* mutants (G2032R, L1951R, S1986F, L2026 and D2033N) with 90 times higher efficacy than crizotinib [106,107]. In June 2017, repotrectinib received FDA orphan drug designation for the treatment of NSCLC harbouring the aforementioned oncogenic mutations. In terms of efficacy, 33 NSCLC patients harbouring *ROS1* rearrangements were recruited in a larger phase I study including patients with different solid malignancies and mutations (e.g., *TKR* and *ALK*) and administered with increasing dosage of repotrectinib [108]. While TKI pretreated patients reported an ORR of 39% with an intracranial response rate of 75%, TKI-naïve patients reported an ORR of 82% and an intracranial response rate of 100%. Importantly, one patient died following repotrectinib administration, but tumour regression was achieved by five patients who had received prior crizotinib therapy. In terms of toxicity, adverse events were observed for the vast majority of patients, with dizziness (57%) and dysgeusia (51%) being the most frequent AEs [108]. The ability to target the ATP-binding site of ROS1 and overcome steric interference created by different mutations meant repotrectinib showed activity against multiple resistance mechanisms, including metastasis, bypass pathways and different mutations.

### 7.7. Other Promising Drugs

It is worth mentioning additional drugs that are thought to be potentially valuable ROS1 inhibitors. Of these, ensartinib is a low molecular weight, orally available agent currently under evaluation for paediatric *ROS1*-rearranged patients [109]. Fortinib, although discontinued since 2015, has been shown to be a highly effective ROS1 inhibitor and demonstrated activity against crizotinib-resistant ROS1 mutations [110].

## 8. Discussion

ROS1 inhibitors have demonstrated clinical efficacy against *ROS1*-rearranged malignancies [75,85,93,102,104,108]. Crizotinib has been a game-changer for the treatment of advanced *ROS1*-positive lung cancers and it is currently the preferred first-line option for these patients [68]. Unfortunately, the vast majority of patients have reported disease progression and resistance to crizotinib. A new generation of agents capable of ROS1 inhibition is therefore required, mainly for more robust delivery of therapy and penetration of the CNS. Preliminary data from ongoing clinical studies assessing novel inhibitor compounds are very promising, and although *ROS1*-rearranged cancers are relatively rare, new therapies might provide clinicians with treatment suggestions. Agents including repotrectinib and lorlatinib have demonstrated significant efficacy in the post-crizotinib setting [104,108]. The previously mentioned homology with *ALK*-rearranged lung cancers can, moreover, improve understanding of *ROS1*-driven NSCLC cases under treatment and drive helpful clinical suggestions [111]. Of the numerous strategies employed by cancer cells to evade inhibition, the most studied is point mutation of drug targets, whereby a second mutation event results in resistance to multiple inhibitors [112]. In future, it will be paramount to detect and verify the different resistance mechanisms employed to allow the avoidance of disease progression in lung cancer patients administered with *ROS1*-targeted therapies [112]. Investigation of bypass signalling pathways, an additional cause of therapy resistance in *ROS1* lung cancer, is essential for generation of superior compounds for combination therapies.

On the clinical side, techniques for early stage identification of resistance mechanisms from tissue or blood would facilitate more precise treatment plans with a higher probability of preventing cancer progression and overcoming drug resistance. In this light, it is worth mentioning the first liquid biopsy effort for detection and monitoring of ROS1 circulating tumour DNA (ctDNA) in *ROS1*-rearranged malignancies [80,113]. We hypothesise that a biopsy at the time of disease progression would enable identification and characterisation of drug resistance mechanisms to permit more informed and precise treatment regimens.

## 9. Conclusions

Its inability to penetrate the BBB renders crizotinib unsuitable for ROS1-rearranged NSCLC patients with CNS metastasis; in such cases, entrectinib, ceritinib, repotrectinib and lorlatinib are more appropriate options since these agents were specifically engineered to penetrate the BBB. With that said, although ceritinib and entrectinib are less effective towards crizotinib-resistant malignancies, they showed increased efficacy towards brain metastasis with entrectinib the current preferred and FDA-approved first-line treatment for those with metastatic brain involvement [46] According to the latest results from various clinical trials, repotrectinib and lorlatinib are the most potent ROS1 inhibitors, with good CNS activity and the ability to overcome resistance resulting from bypass signalling activation and alternative *ROS1* mutations. Of note, repotrectinib is currently under investigation while lorlatinib has recently been approved for *ALK*-rearranged NSCLC patients and is the current preferred second-line treatment in case of no evidence of G2032R mutation and/or bypass signalling [114]. To summarise, early detection of *ROS1*-rearranged lung cancers is essential as there are several targeted therapies available, despite the low incidence rate. The understanding of underlying molecular alterations that lead to drug resistance and disease progression, and the development of novel drugs to counteract this, is essential for the design of more precise and effective combination therapies.

## Figures and Tables

**Figure 1 cancers-12-03293-f001:**
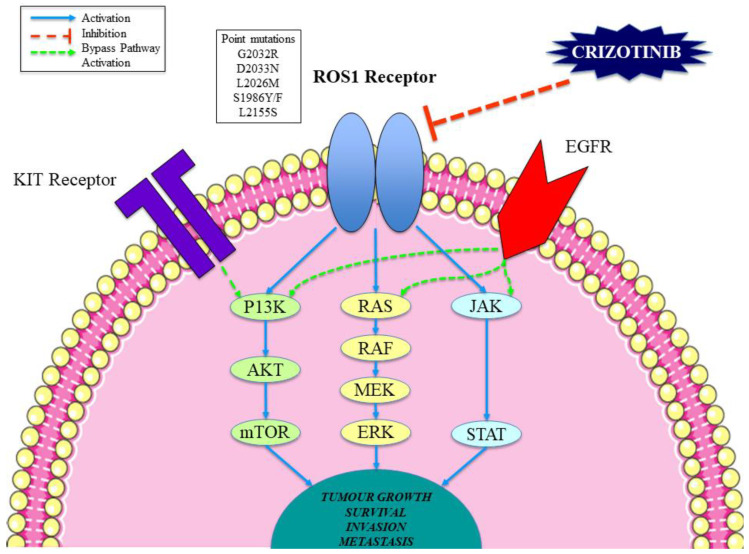
Molecular mechanisms of crizotinib in ROS1-rearranged lung cancer patients.

**Table 1 cancers-12-03293-t001:** Main *ROS1* fusion partners in non-small cell lung cancer (NSCLC).

No	Gene	Alias	Description	Estimated Frequency in NSCLC	Reference
1	*CD74*		Cluster of differentiation 74	32–42%	[13]
2	*SLC34A2*		Solute carrier family 34 member 2 gene	12–18%	[13]
3	*EZR*		Ezrin gene	6–15%	[2]
4	*TMP3*		Tropomyosin 3 gene	3–15%	[2]
5	*SDC4*		Syndecan 4 gene	7–11%	[2]
6	*FIG*	*GOPC*	Fused in Glioblastoma	2–3%	[11]
7	*TMEM106B*		Transmembrane protein 106B	1%	[11]
8	*CCDC6*		Coiled-coil domain containing 6 gene	1%	[14]
9	*LIMA1*	*EPLIN*	LIM domain and actin-binding 1 gene	1%	[5]
10	*WNK1*		Lysine deficient protein kinase 1	1%	[20]
11	*LRIG3*		Leucine-rich repeats and immunoglobulin-like domains 3 gene	1%	[2]
12	*TDP52L1*		Tumour protein D52 like 1 gene	1%	[11]
13	*CLTC*		Clathrin heavy chain gene	1%	[16]
14	*MSN*		Moesin gene	1%	[5]
15	*KDELR2*	*ELP-1* *ERD2.2*	Endoplasmic reticulum protein retention receptor 2 gene	1%	[14]
16	*MYO5C*		Myosin VA (heavy chain 12, myoxin)	1%	[19]
17	*TFG*		TRK-fused gene	1%	[23]
18	*RBPMS*		RNA-binding protein with multiple splicing	1%	[21]

**Table 2 cancers-12-03293-t002:** Ongoing clinical trials using TKI against ROS1-rearranged lung cancer.

Clinical Trial Identifier	Study Design	Intervention/s	Setting	Primary Endpoint	Phase	Status
NCT03399487	46 Participants, Single group assignment, Non-Randomized, Open label	LDK378 (Ceritinib)	Second line	ORR	2	Recruiting
NCT03972189	111 Participants, Single group assignment, Non-Randomized, Open label	TQ-B3101	Second line	ORR	2	Recruiting
NCT02927340	30 Participants, Single group assignment, Non-Randomized, Open label	Lorlatinib	First or later line	DCR	2	Recruiting
NCT01639508	68 Participants, Single group assignment, Non-Randomized, Open label	Cabozantinib	Second line	ORR	2	Recruiting
NCT01970865	334 Participants, Non-Randomized, Open label	PF-06463922 Crizotinib	First or later line	DLT (phase 1) OR (phase 2)	2	Active, not recruiting
NCT04302025	60 Participants, Single group assignment, Non-Randomized, Open label	Alectinib Entrectinib Vemurafenib Cobimetinib	Second or later line	MPR	2	Not yet recruiting
NCT04084717	50 Participants, Parallel assignment, Non-Randomized, Open label	Crizotinib	Second line	RR	2	Not yet recruiting
NCT03088930	18 Participants, Single group assignment, Non-Randomized, Open label	Crizotinib	Second line	RR	2	Recruiting
NCT03087448	69 Participants, Single group assignment, Non-Randomized, Open label	Ceritinib (Phase 1) Trametinib (Phase 2)	Second or later line	MTD	1–2	Recruiting
NCT02183870	30 Participants, Single group assignment, Non-Randomized, Open label	Crizotinib	Any prior treatment	ORR	2	Active, not recruiting
NCT01964157	32 Participants Single group assignment Non-randomized Open label	LDK378	Second or later line	ORR	2	Recruiting
NCT04292119	96 Participants Randomised Parallel Assignment Open label	Lorlatinib, Binimetinib, Crizotinib	Any prior treatment	MTD, OR	1–2	Recruiting
NCT03608007	69 Participants Single group assignment Non-randomised Open label	X-396 Capsule (Ensartinib)	Second line	OR	2	Recruiting
NCT03718117	70 Participants Cohort, Prospective	Crizotinib	Any prior treatment	Demographics	/	Active, not recruiting
NCT04005144	18 Participants Non-randomized Single group assignment Open label	Brigatinib + Binimetinib	Second-line or later	AE, DLT	1	Recruiting
NCT02568267	300 Participants Non-randomised Parallel Assignment Open label	Entrectinib	Any prior treatment	OR	2	Recruiting

**Table 3 cancers-12-03293-t003:** Techniques for the detection of ROS1 rearrangements.

Technique	Advantages	Disadvantages
IHC	High sensitivity Economic Easy to use Short turnaround Low number of cells required Employed on conventional AFC	Subjective evaluation Tissue fixation procedure Antibody chemical properties
FISH	Long time trustworthiness Reliability “Gold standard” tool	Moderately expansive Difficult to use Labour intense False-negative results
RT-PCR	Good specificity Good sensitivity Low amount of starting material	FFPE samples only Error-prone More validation required Good quality of starting material
NGS	Detection of novel fusions Parallel identification FFPE or biopsy specimen	More validation required Turnaround time Specimen depletion

IHC: Immunohistochemistry; FISH: Fluorescent in-situ hybridization; PCR: polymerase chain reaction; NGS: next-generation sequencing; AFC: alcohol-fixed cytology; FFPE: formalin-fixed and formalin-embedded.

**Table 4 cancers-12-03293-t004:** *ROS1* mutations conferring resistance to crizotinib and mechanism of action.

Mutation	ROS1 fusion	Location	Mechanism
D2033N	CD74-ROS1	Kinase hinge	Modification of electrostatic forces
G2032R	CD74-ROS1	Kinase hinge	Steric interference
L2026M	CD74-ROS1	Inhibitor binding site	Hindrance of drug binding
L2155S	SLC34A2-ROS1	Not known	Protein malfunction
S1986F/Y	EZR-ROS1	Not known	Obstruction of the active site of the enzyme

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
