# Peer review of "Focus on ROS1-Positive Non-Small Cell Lung Cancer (NSCLC): Crizotinib, Resistance Mechanisms and the Newer Generation of Targeted Therapies"

_cancers, 2020, doi:10.3390/cancers12113293_

Round 1
Reviewer 1 Report
This paper on Crizotinib resistance in ROS1-positive NSCLC is a review that summarizes the available data in a comprehensive and exhaustive manner. As for my part, I have only one remark, which I would recommend to be taken into consideration. It refers to lines 120 to 124, which deal with whole-brain irradiation versus stereotactic irradiation.
I would rephrase the following sentence: “Alternatively, when fewer lesions are observed (three or four), stereotactic radiation is preferred (also known as gamma knife), which brings fewer side effects but higher chances of in-situ relapse.“
- Stereotactic radiation is a term that comprises three types of technically different approaches: LINAC-based stereotactic radiotherapy, gammaknife and cyberknife. The clinical outcome in all three of them is basically the same.
- Stereotactic approaches yield excellent local control rates comparable to neurosurgical techniques, therefore the notion of higher chances of in-situ relapse is wrong. The risk of in-brain relapses is higher with stereotactic or neurosurgical approaches alone than with a combination of these highly focal therapies and whole brain irradiation (e.g. Andrews Lancet 2004, Martin Kocher JCO 2011).
Author Response
Dear Reviewer,
We are very grateful for your time and effort in reading and commenting on our manuscript. Please find below our responses.
Reviewer #1
I would rephrase the following sentence: “Alternatively, when fewer lesions are observed (three or four), stereotactic radiation is preferred (also known as gamma knife), which brings fewer side effects but higher chances of in-situ relapse.“
- Stereotactic radiation is a term that comprises three types of technically different approaches: LINAC-based stereotactic radiotherapy, gammaknife and cyberknife. The clinical outcome in all three of them is basically the same.
- Stereotactic approaches yield excellent local control rates comparable to neurosurgical techniques, therefore the notion of higher chances of in-situ relapse is wrong. The risk of in-brain relapses is higher with stereotactic or neurosurgical approaches alone than with a combination of these highly focal therapies and whole brain irradiation (e.g. Andrews Lancet 2004, Martin Kocher JCO 2011).
Thank you for raising this point. We have rephrased the sentence accordingly and included the suggested references within our bibliography.
Reviewer 2 Report
This review addresses a specific subset of non-small cell lung cancers harbouring ROS1 gene rearrangements, discussing aspects such as treatment outcome and resistance mechanisms of current and newer generations of tyrosine kinase inhibitors targeting ROS1. The authors describe the prevalence of ROS1 genetic alterations in NSCLC, the front-line chemotherapy approach and treatment outcome. Finally, a discussion on the performance of various tyrosine kinase inhibitors on ongoing clinical trials is provided.
In section 3. ROS1 in lung cancer and brain metastasis, the authors report that 1-2% of NCLC patients harbour ROS1 gene rearrangements which co-occur with other genetic aberrations. To overview the different co-occurrences, treatment regimens and outcomes, the authors could provide the resumed information on a scheme/table depicting gene mutations/rearrangements together with its frequencies and treatment outcome.
Author Response
Dear Reviewer,
We are very grateful for your time and effort in reading and commenting on our manuscript. Please find below our responses.
Reviewer #2
In section 3. ROS1 in lung cancer and brain metastasis, the authors report that 1-2% of NCLC patients harbour ROS1 gene rearrangements which co-occur with other genetic aberrations. To overview the different co-occurrences, treatment regimens and outcomes, the authors could provide the resumed information on a scheme/table depicting gene mutations/rearrangements together with its frequencies and treatment outcome.
Thank you for the suggestion. We have implemented Table 1 with estimated relative frequencies for each ROS1 fusion partner. Unfortunately, due to the paucity of patients – most of the fusion genes have been reported once only – data on survival and treatment outcomes are lacking. Survival analysis for ROS1 fusions has only been rarely addressed and is still controversial.
Reviewer 3 Report
Some suggestions:
1) Entrectinib is also FDA approved.
2) KRAS G12 C is being evaluated in a Phase I trial and showed promising activity in a heavily pretreated population (NEJM 2020).
3) Brain metastases: Entrectinib is currently approved and is the preferred TKI for patients with brain metastases.
4) Lorlatinib is not FDA approved for ROS-1.
5) Conclusion: again, entrectinib is the preferred 1st line TKI with brain metastases (FDA approved). Lorlatinib is the preferred 2nd line TKI if no evidence of G2032R, histologic transformation or bypass signaling.
Author Response
Dear Reviewer,
We are very grateful for your time and effort in reading and commenting on our manuscript. Please find below our responses.
Reviewer #3
Some suggestions:
1) Entrectinib is also FDA approved.
Thank you for raising this issue. We have added this information throughout the manuscript and also commented on it.
2) KRAS G12 C is being evaluated in a Phase I trial and showed promising activity in a heavily pretreated population (NEJM 2020).
Thank you for the suggestion. We amended the manuscript accordingly.
3) Brain metastases: Entrectinib is currently approved and is the preferred TKI for patients with brain metastases.
Thank you for this suggestion. We amended the manuscript accordingly.
4) Lorlatinib is not FDA approved for ROS-1.
Thank you for pointing out this issue. We modified the information accordingly.
5) Conclusion: again, entrectinib is the preferred 1st line TKI with brain metastases (FDA approved). Lorlatinib is the preferred 2nd line TKI if no evidence of G2032R, histologic transformation or bypass signaling.
Thank you for the suggestion. We have amended the “Conclusions” section accordingly and properly referenced the amended paragraph.
Round 2
Reviewer 2 Report
The manuscript has suffered some improvements and the authors efforts are appreciated.
The authors have included a table with the main ROS1 fusion partners in NSCLC (Table 1). Currently the reported number of ROS1 fusion genes is eighteen. Since recently a similar review has been published an effort to include the latest information in the text would beneficiate the impact of the manuscript.
The authors have referenced Figure 1 which lacks a descriptive figure legend.
Author Response
The manuscript has suffered some improvements and the authors efforts are appreciated.
Thank you for your positive feedback.
The authors have included a table with the main ROS1 fusion partners in NSCLC (Table 1). Currently, the reported number of ROS1 fusion genes is eighteen. Since recently a similar review has been published an effort to include the latest information in the text would beneficiate the impact of the manuscript.
Thank you for this suggestion. We have implemented Table 1 with missing ROS1 fusion partners including the RBPMS fusion gene partner which has been reported earlier this year with according references. Lastly, we created a column reporting alternative gene names (“alias”) to avoid any misinterpretation.
The authors have referenced Figure 1 which lacks a descriptive figure legend.
Thank you for raising this point. We have added figure caption within the comments column, aside of the figure.